# Learning Better Structured Representations Using Low-rank Adaptive Label Smoothing

**Asish Ghoshal, Xilun Chen, Sonal Gupta, Luke Zettlemoyer & Yashar Mehdad**
`{aghoshal,xilun,sonalgupta,lsz,mehdad}@fb.com`
Facebook AI

## Abstract

Training with soft targets instead of hard targets has been shown to improve performance and calibration of deep neural networks. Label smoothing is a popular way of computing soft targets, where one-hot encoding of a class is smoothed with a uniform distribution. Owing to its simplicity, label smoothing has found widespread use for training deep neural networks on a wide variety of tasks, ranging from image and text classification to machine translation and semantic parsing. Complementing recent empirical justification for label smoothing, we obtain PAC-Bayesian generalization bounds for label smoothing and show that the generalization error depends on the choice of the noise (smoothing) distribution. Then we propose low-rank adaptive label smoothing (LORAS): a simple yet novel method for training with *learned* soft targets that generalizes label smoothing and adapts to the latent structure of the label space in structured prediction tasks. Specifically, we evaluate our method on semantic parsing tasks and show that training with appropriately smoothed soft targets can significantly improve accuracy and model calibration, especially in low-resource settings. Used in conjunction with pre-trained sequence-to-sequence models, our method achieves state of the art performance on four semantic parsing data sets. LORAS can be used with any model, improves performance and implicit model calibration without increasing the number of model parameters, and can be scaled to problems with large label spaces containing tens of thousands of labels.

## 1 Introduction

Ever since Szegedy et al. (2016) introduced *label smoothing* as a way to regularize the classification (or output) layer of a deep neural network, it has been used across a wide range of tasks from image classification (Szegedy et al., 2016) and machine translation (Vaswani et al., 2017) to pre-training for natural language generation (Lewis et al., 2019). Label smoothing works by mixing the one-hot encoding of a class with a uniform distribution and then computing the cross-entropy with respect to the model's estimate of the class probabilities to compute the loss of the model. This prevents the model being too confident about its predictions — since the model is now penalized (to a small amount) even for predicting the correct class in the training data. As a result, label smoothing has been shown to improve generalization across a wide range of tasks (Müller et al., 2019). More recently, Müller et al. (2019) further provided important empirical insights into label smoothing by showing that it encourages the representation learned by a neural network for different classes to be equidistant from each other.

Yet, label smoothing is overly crude for many tasks where there is structure in the label space. For instance, consider task-oriented semantic parsing where the goal is to predict a parse tree of intents, slots, and slot values given a natural language utterance. The label space comprises of ontology (intents and slots) and natural language tokens and the output has specific structure, e.g., the first token is always a top-level intent (see Figure 1), the leaf nodes are always natural language tokens and so on. Therefore, it is more likely for a well trained model to confuse a top-level intent with another top-level intent rather than a natural language token. This calls for models whose uncertainty is spread over related tokens rather than over obviously unrelated tokens. This is especially important in the few-shot setting where there are few labelled examples to learn representations of novel tokens from.

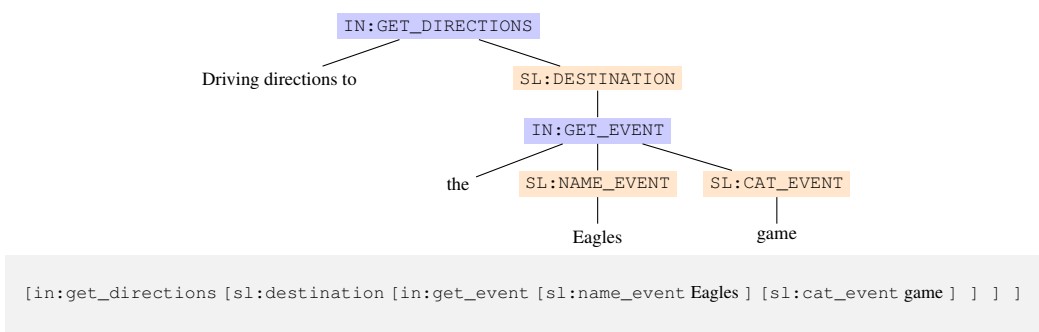

Figure 1: **Top**: Semantic parse tree of the utterance "Driving directions to the Eagles game". **Bottom:** Serialized tree. IN: stands for *intents* while SL: stands for *slots* (See Gupta et al., 2018).

**Our contributions.** We present the first rigorous theoretical analysis of label smoothing by obtaining PAC Bayesian generalization bounds for a closely related (upper-bound) loss function. Our analysis reveals that the choice of the smoothing distribution affects generalization, and provides a recipe for tuning the smoothing parameter.

Then, we develop a simple yet effective extension of label smoothing: *low-rank adaptive label smoothing* (LORAS), which provably generalizes the former and adapts to the latent structure that is often present in the label space in many structured prediction problems. We evaluate LORAS on three semantic parsing data sets, and a semantic parsing based question-answering data set, using various pre-trained representations like RoBERTa Liu et al. (2019) and BART Lewis et al. (2019). On ATIS (Price, 1990) and SNIPS (Coucke et al., 2018), LORAS achieves average absolute improvement of 0.6% and 0.9% respectively in exact match of logical form over vanilla label smoothing across different pre-trained representations. In the few-shot setting using the TOPv2 data set (Chen et al., 2020)[1], LORAS achieves an accuracy of 74.1% on average over the two target domains — an absolute improvement of 2% over using vanilla label smoothing and matching the state-of-the-art performance in Chen et al. (2020) despite their use of a much more complex meta-learning method. Lastly, in the transfer learning setting on the Overnight data set (Wang et al., 2015), LORAS improves over vanilla label smoothing by 1% on average on the target domains. Furthermore, LORAS is easy to implement and train and can be used in conjunction with any architecture.

We show that unlike vanilla label smoothing, LORAS effectively solves the neural network overconfidence problem for structured outputs where it produces more calibrated uncertainty estimates over different parts of the structured output. As a result, LORAS reduces the test set expected calibration error by 55% over vanilla label smoothing on the TOPv2 data set.

We present an efficient formulation of LORAS which does not increase the model size, while requiring only $\mathcal{O}(K)$ additional memory during training where $K$ is the output vocabulary size (or the number of classes in the multi-class setting).

## 2 PRELIMINARIES

We consider structured prediction formulated as a sequence-to-sequence (*seq2seq*) prediction problem. We motivate our method through semantic parsing where the input $x$ is an natural language utterance and the output $y$ is a serialized tree that captures the semantics of the input in a machine understandable form (see Figure 1 for an example). Specifically, given input output pairs $(x, y)$ where $x = (x_i)_{i=1}^m$ and $y = (y_i)_{i=1}^n$ are sequences, let $\phi(x, y_{1:t-1})$ be the representation of the input and output tokens up to time step $t - 1$ modeled by a neural network. At time step $t$ the probability of the $t$-th output token is given by: $\mathrm{softmax}(\mathbf{W}\phi(x, y_{1:t-1}))$, where $\mathbf{W} \in \mathbb{R}^{K \times d}$ are the output projection weights (last layer) of the neural network and $K$ is the vocabulary size. The representation and the output projections are learned by minimizing the negative log-likelihood of the observed samples $S$.

---

[1] TOPv2 data set is a newer version of the TOP data set introduced in (Gupta et al., 2018) containing 6 additional domains, which is particularly suitable for benchmarking few-shot semantic parsing methods.

**Label Smoothing.** The idea behind label smoothing is to uniformly smooth the one-hot vector before computing the cross-entropy with the learned distribution. Let $\mathbf{y}_t = (\mathbf{1}\left[y_t = j\right])_{j=1}^K$ denote the one-hot encoding of $t$-th output token and $\mathbf{p}_t = (p(y_t = j \mid x, y_{1:t-1}))_{j=1}^K$ denote the distribution over the vocabulary modeled by the neural network. Then, setting $\mathbf{y}_t^{LS} = (1 - \alpha)\mathbf{y} + \alpha^1/K$, we compute $H(\mathbf{y}_t^{LS}, \mathbf{p}_t)$, the cross-entropy between $\mathbf{y}_t^{LS}$ and $\mathbf{p}_t$, as our loss function:

$$H(\mathbf{y}_t^{LS}, \mathbf{p}_t) = -(1 - \alpha) \sum_{j=1}^K \mathbf{y}_{t,j} \log \mathbf{p}_{t,j} - \frac{\alpha}{K} \sum_{j=1}^K \log \mathbf{p}_{t,j}. \tag{1}$$

## 3 THEORETICAL MOTIVATION FOR LABEL SMOOTHING

In this section we look at why training neural networks with soft-targets can help with generalization. To simplify exposition we will consider a multi-class classification setting where we have input output pairs $(x, y)$ and $y \in [K]$. As first described by Müller et al. (2019), label smoothing encourages the representation of the input $\phi(x)$ to be close to projection weight $(\mathbf{w})$ for the correct class and at the same time be equidistant from the weights of all other classes. We will formalize this by obtaining rigorous generalization bounds for label smoothing. Towards that end we will fix the input representation $\phi(x) \in \mathbb{R}^d$ with $\|\phi(x)\|_2 \leq 1$ and focus on the classification layer weights $\mathbf{W} \in \mathbb{R}^{K \times d}$. For a noise distribution $n(\cdot)$, which is uniform for standard label smoothing, an upper-bound on the loss (1) is given as follows:

$$\widetilde{L}(S, \mathbf{W}) = \sum_{(x,y) \in S} l(x, y; \mathbf{W}, \alpha) + \frac{\alpha}{2} \left\| \mathbf{n} - \mathbf{W}\overline{\phi} \right\|_2^2 \tag{2}$$

where $\mathbf{n} = (n(i))_{i=1}^K$ is the vectorized noise distribution and $\overline{\phi} = \sum_{x \in S} \phi(x)$ is the sum of input representations in the training set, and $l(x, y; \mathbf{W}, \alpha)$ is the re-scaled negative log-likelihood of the observed data $S$ where the linear term is scaled by $\alpha$.

The upper bound is obtained by ignoring the norm of $\mathbf{W}\overline{\phi}$ from the objective (see Appendix A.1 for a derivation). The objective given by (2) is essentially penalized negative log-likelihood with a penalty term that encourages the aggregated (un-normalized) class scores to be close to the noise distribution. Unlike standard weight regularization, however, the regularization term depends on both the weight $\mathbf{W}$ and the inputs $x \in S$. Therefore, existing theory for regularized loss minimization doesn't apply to this case and we invoke PAC-Bayesian theory to analyze the above. As is standard in PAC-Bayesian analysis, we consider minimizing the empirical loss $\widetilde{L}(S, \mathbf{W})$ around a small neighborhood of the weight $\mathbf{W}$. For a posterior distribution $Q$ on the weights $\mathbf{W}$, which depends on the sample $S$, we consider the following empirical risk and the expected risk and their respective minimizers.

$$\widehat{R}(Q(\mathbf{W})) = \mathbb{E}_{\mathbf{W}' \sim Q(\mathbf{W})} \left[ \widetilde{L}(S, \mathbf{W}) \right] \qquad \text{(empirical risk)}$$

$$\overline{R}(Q(\mathbf{W})) = \mathbb{E}_{\mathbf{W}' \sim Q(\mathbf{W})} \left[ \mathbb{E}_{(x,y)} \left[ l(x, y; \mathbf{W}', \alpha) \right] \right] \qquad \text{(expected risk)}$$

$$\widehat{\mathbf{W}} \in \underset{W \in \mathbb{R}^{K \times d}}{\operatorname{argmin}} \widehat{R}(Q(\mathbf{W})) \qquad \text{(empirical minimizer)}$$

$$\overline{\mathbf{W}} \in \underset{W \in \mathbb{R}^{K \times d}}{\operatorname{argmin}} \overline{R}(Q(\mathbf{W})) \qquad \text{(true minimizer)}$$

The following theorem, whose proof we defer to Appendix A.2, bounds the risk of the minimizer of the label smoothing loss (on the sample $S$) in terms of the risk of the minimizer of the expected negative log-likelihood (under the data distribution).

**Theorem 1** (PAC-Bayesian generalization bound). *Set the distribution $Q(\mathbf{W})$, parameterized by $\mathbf{W}$ with bounded induced norm, over the weights $\mathbf{W}'$ to be such that each column $\mathbf{W}'_{*,i}$ is sampled i.i.d. from the Gaussian distribution $\mathcal{N}(\mathbf{W}\overline{\phi}, \mathbf{I})$. If $\alpha = 2d/\sqrt{N}$, where $N = |S|$ is the number of samples, then with probability at least $1 - \delta$ the generalization error is bounded as follows:*

$$\overline{R}(Q(\widehat{\mathbf{W}})) - \overline{R}(Q(\overline{\mathbf{W}})) \leq \frac{2d}{\sqrt{N}} \left\| \mathbf{n} - \overline{\mathbf{W}}\overline{\phi} \right\|_2^2 + \frac{1}{\sqrt{N}} \log \frac{2e^{\frac{b^2}{8}}}{\delta}.$$

It is important to note that the generalization error depends on the number of classes $K$ through the term $\left\|\mathbf{n} - \overline{\mathbf{W}}\overline{\phi}\right\|_2$ which grows as $\Theta(K)$. This is due to the fact that the label smoothing objective regularizes the class scores ($\mathbf{W}\overline{\phi}$) as opposed to regularizing the output layer weights ($\mathbf{W}$) directly which would result in the generalization error depending on $\|\mathbf{W}\|_F^2$ which is $\Theta(dK)$. The above result also prescribes how to set the smoothing constant $\alpha$ which is typically chosen to be the constant $0.1$ in practice — as the number of samples $N \to \infty$, $\alpha \to 0$ and less smoothing of the hard targets is needed to achieve generalization. Furthermore, the generalization error also depends on how close the aggregated un-normalized class scores ($\overline{\mathbf{W}}\overline{\phi}$) of the true minimizer on the training set $S$ are to the noise distribution. Therefore, choosing a more informative smoothing distribution, as opposed to a uniform distribution, should lead to better generalization.

## 4 Low-Rank Adaptive Label Smoothing

Motivated by the above result, we propose to use a more informative noise distribution than the uniform distribution to smooth the hard targets. The idea behind low-rank adaptive label smoothing (LORAS) is to learn the noise distribution that is to be mixed in with the one-hot target jointly with the model parameters. Specifically, we consider noise distributions $n(\cdot \mid y, \mathbf{S})$ parameterized by the true label $y$ and a symmetric matrix $\mathbf{S} \in \mathbb{R}^{K \times K}$. Then, in LORAS we set $\mathbf{y}_t^{\text{LORAS}} = (1 - \alpha)\mathbf{y}_t + \alpha\mathbf{n}(y_t, \mathbf{S})$ where $\mathbf{n}(y_t, \mathbf{S}) = \text{softmax}(\mathbf{y}_t\mathbf{S})$ and compute the cross-entropy as follows:

$$H(\mathbf{y}_t^{\text{LORAS}}, \mathbf{p}_t) = -\sum_{j=1}^{K} \mathbf{y}_t^{\text{LORAS}} \log \mathbf{p}_t$$

$$= -(1 - \alpha)\sum_{j=1}^{K} \mathbf{y}_{t,j} \log \mathbf{p}_{t,j} - \alpha \sum_{j=1}^{K} \frac{\exp(\mathbf{S}_{y_t,j})}{Z(\mathbf{S}_{y_t,*})} \log \mathbf{p}_{t,j}, \qquad (3)$$

where $\mathbf{S}_{y_t,*}$ denotes the $y_t$-th row of the matrix $\mathbf{S}$ and $Z(\mathbf{S}_{y_t,*}) = \sum_{l=1}^{K} \exp(\mathbf{S}_{y_t,l})$ is the partition function. Thinking of $\mathbf{n}_j$ for all $j \in [K]$ as Lagrange multipliers, the matrix $\mathbf{S}$ serves to relax the constraint that the representation of an output token be equally close to the weight vectors for all other tokens. For instance, in the semantic parsing task where the vocabulary comprises of ontology tokens (intents and slots) and utterance tokens, it is unreasonable to treat everything on equal footing since certain groups of tokens (e.g. ontology tokens) might more similar to each other than others. For the $i$-th output token $\mathbf{S}_{i,j}$ determines "how close" the representation of the $i$-th token should be to the weight $\mathbf{w}_j$ of the $j$-th token, with larger (resp. smaller) values of $\mathbf{S}_{i,j}$ forcing the representation to be closer (resp. farther). With this interpretation, the matrix $\mathbf{S}$ can also be thought of as imposing a notion of similarity over vocabulary tokens with $\mathbf{S}_{i,j}$ determining how similar the $i$-th token is to the $j$-th token.

**Low-rank assumption.** The size of the vocabulary in many NLP applications is often large containing tens of thousands of tokens. Since the size of $\mathbf{S}$ is quadratic in the size of the vocabulary, it is prohibitive to work the matrix $\mathbf{S}$ directly. Instead, we assume that $\mathbf{S}$ has low rank structure, i.e., there exists an $\mathbf{L} \in \mathbb{R}^{K \times r}$ with $r \ll K$ such that $\mathbf{S} = \mathbf{L}\mathbf{L}^\top$. This is a reasonable assumption in many settings, especially in our semantic parsing application, where a natural group structure exists in the vocabulary comprising of ontology and utterance tokens. Next, we formulate (3) in terms of $\mathbf{L}$ thereby eschewing the matrix $\mathbf{S}$ altogether. Under the above assumption we have:

$$\mathbf{y}_t^{\text{LORAS}} = (1 - \alpha)\mathbf{y}_t + \alpha\text{softmax}(\mathbf{L}_{y_t,*}\mathbf{L}^\top).$$

Note that $\mathbf{L}_{y_t,*}\mathbf{L}^\top \in \mathbb{R}^K$ and we don't need to explicitly construct the matrix $\mathbf{S}$ which can be large.

**Entropy constraint and dropout.** To encourage discovery of latent structure in the label space, we minimize $H(\mathbf{y}_t^{\text{LORAS}}, \mathbf{p}_t)$ subject to an entropy constraint on $\text{softmax}(\mathbf{S}_{k,*})$. This forces the noise distribution to be farther away from the uniform distribution. Also note that the low-rank assumption ensures that $\mathbf{S}$ doesn't become close to a diagonal matrix which would reduce adaptive label smoothing to the case of training with hard targets. To see this, assume that for some $i$ $\mathbf{S}_{i,i} \gg \mathbf{S}_{i,j} \; \forall j \neq i$. Then $\text{softmax}(\mathbf{S}_{i,*}) \approx \mathbf{e}_i$, where $\mathbf{e}_i$ is a vector of all zeros except for a 1 at the $i$-th index, thereby reducing adaptive label smoothing to no label smoothing. The low rank

assumption ensures that matrix $\mathbf{S}$ does not become diagonally dominant. Lastly, to avoid the matrix $\mathbf{L}$ from overfitting to the training data, we add dropout on $\mathbf{L}$. The final objective that adaptive label smoothing optimizes is given in the following, where $H(\cdot, \cdot)$ denotes cross-entropy while $H(\cdot)$ denotes entropy:

$$\widetilde{\mathbf{L}} = dropout(\mathbf{L}; q)$$

$$L^{\text{LORAS}}(S) = \sum_{(x,y)\in S} \sum_{t=1}^{|y|} H(\mathbf{y}_t^{\text{LORAS}}, \mathbf{p}_t) + \eta H(\text{softmax}(\widetilde{\mathbf{L}}_{y_t,*}\widetilde{\mathbf{L}}^{\top}))$$

$$= \sum_{(x,y)\in S} \sum_{t=1}^{|y|} H(\mathbf{y}_t^{\text{LORAS}}, \mathbf{p}_t) - \eta (\widetilde{\mathbf{L}}_{y_t,*}\widetilde{\mathbf{L}}^{\top})^{\top} \text{softmax}(\widetilde{\mathbf{L}}_{y_t,*}\widetilde{\mathbf{L}}^{\top})$$

$$+ \eta \, logsumexp(\widetilde{\mathbf{L}}_{y_t,*}\widetilde{\mathbf{L}}^{\top}), \qquad (4)$$

where

$$H(\mathbf{y}_t^{\text{LORAS}}, \mathbf{p}_t) = -(1-\alpha)\sum_{j=1}^{K} \mathbf{y}_{t,j} \log \mathbf{p}_{t,j} - \alpha \sum_{j=1}^{K} \frac{\exp(\widetilde{\mathbf{L}}_{y_t,*}(\widetilde{\mathbf{L}}_{j,*})^{\top})}{Z(\widetilde{\mathbf{L}}_{y_t,*}\widetilde{\mathbf{L}}^{\top})} \log \mathbf{p}_{t,j}, \qquad (5)$$

$\eta \geq 0$ is a hyper-parameter that controls how far the noise distributions are from the uniform distribution, with larger values encouraging more peaked (low-entropy) noise distributions, and $q \in [0, 1)$ is the dropout parameter. The matrix $\mathbf{L}$ is initialized to be a matrix of all ones, and we jointly learn the matrix $\mathbf{L}$ along with the model parameters. After training, the matrix $\mathbf{L}$ is discarded. Next, we show that the above formulation of adaptive label smoothing strictly generalizes label smoothing.

**Proposition 1.** *Setting* $q = \eta = 0$ *and* $\mathbf{L} = \mathbf{1}$, *where* $\mathbf{1}$ *is the* $K$-*dimensional vector of ones,* $L^{\text{LORAS}}(S) = L^{LS}(S) = \sum_{(x,y)\in S}\sum_{t=1}^{|y|} H(\mathbf{y}_t^{LS}, \mathbf{p}_t)$.

Since setting $q = \eta = 0$, and the rank parameter $r = 1$, we have that $\mathbf{L} = \mathbf{1}$ is in the solution path of minimizing the LORAS loss given in (4). Therefore, adaptive label smoothing strictly generalizes label smoothing.

## 5 EXPERIMENTS

We evaluate LORAS on three semantic parsing data sets: ATIS (Price, 1990), SNIPS (Coucke et al., 2018), and TOPv2, (Chen et al., 2020), and a question answering data set: Overnight (Wang et al., 2015) — where the goal is to predict a executable logical form that can be executed against a database to answer a natural language query. On TOPv2 we evaluate LORAS in the few-shot setting as in (Chen et al., 2020) and for Overnight we consider a transfer learning setting which is detailed in the next paragraph. All data sets were pre-processed exactly as in (Chen et al., 2020). Similar to Chen et al. (2020) we use the state-of-the-art model from (Rongali et al., 2020) as our main model. While for ATIS, SNIPS, and TOPv2 we use *ontology only generation* where the model is only allowed to generate ontology tokens while copying everything else from the utterance, for Overnight data set we place no such constraints on the model since many of the output tokens are neither part of the ontology nor are part of the utterance. We experiment with two different pre-trained language models for initialization: (encoder only) RoBERTa (Liu et al., 2019) and (encoder and decoder) BART (Lewis et al., 2019). For Overnight we only use BART.

We compare the performance of the model with different pre-trained representations with no label smoothing, vanilla label smoothing, and LORAS. For vanilla label smoothing we experiment with $\alpha \in \{0.1, 0.2, 0.3\}$ and report the best accuracy. An $\alpha$ value of 0.1 is typically used in the literature, while an value of more than 0.2 produces inferior results. For LORAS, we experiment with $\alpha \in \{0.1, 0.2, 0.3\}$, a few different rank and dropout parameters, set $\eta = 0.1$ for all the experiments and report the best accuracy. We found that for BART, LORAS dropout parameter of 0.5 worked best, while for RoBERTa dropout of 0.6 worked best. For all three data sets a rank parameter of 25 worked best for LORAS. We did not scale beyond rank 25 since the BART model is fairly memory intensive leaving little room to use large label embedding matrices during training.

| | ATIS | | | | | | SNIPS | | | | | |
|---|---|---|---|---|---|---|---|---|---|---|---|---|
| **Pre-training** | **No LS** | | **LS** | | **LORAS** | | **No LS** | | **LS** | | **LORAS** | |
| | FA | SA | FA | SA | FA | SA | FA | SA | FA | SA | FA | SA |
| RoBERTa | 84.8 | **88.6** | 84.5 | 88.5 | **85.2** | 88.0 | 85.1 | 93.3 | 85.7 | 92.3 | **86.3** | **93.6** |
| BART | 87.6 | **88.6** | 87.7 | 88.5 | **88.2** | **89.8** | 90.0 | 98.0 | 89.4 | 98.0 | **90.6** | **98.3** |
| BERT* | 87.5 | - | 87.9 | - | **88.0** | - | 90.0 | - | 92 | - | **92.3** | - |

Table 1: Test set frame accuracy (FA) and semantic accuracy (SA) on ATIS and SNIPS data set of our seq2seq models with copying mechanism. Numbers in bold represent the best performing method for a given metric and pre-trained model. *Refers to the BERT based joint intent classification and slot tagging model of Chen et al. (2019) where frame accuracy is the sentence level exact match accuracy.

**Few-shot and transfer-learning setting.** In addition to the standard supervised setting, we also evaluate LORAS on the few-shot and transfer learning setting in order to verify the hypothesis that having a more informed smoothing distribution would help when training data is limited. In particular, for TOPv2 we follow the same few-shot domain adaptation setting as in Chen et al. (2020), where 6 domains in the TOPv2 data set are used as source domains while the remaining 2 (*reminder* and *weather*) serve as target domains. For Overnight we consider the four largest domains as source domains and the four smallest domains as target domains. For Overnight, the target domain training set was about one third the size of the source domain training set. We first train our model on the combined training data from source domains and then fine-tune the model on combined training data from target domains. For TOPv2 the training and validation data are limited to 25 samples per intent and slot (25 SPIS) for each target domain. More details about our experimental setup and other data set and training details can be found in Appendix A.4.

## 6 RESULTS

We use the exact match accuracy, referred to as **frame accuracy (FA)** — where the predicted sequence is matched exactly with the target sequence — as the primary evaluation metric following common practice in literature (Rongali et al., 2020). In addition to FA, we also report the **semantic accuracy (SA)**, where we remove the slot values and only compare the resulting target and predicted sequence. For instance, if the target sequence is: [in:playmusic [sl:year eighties ] [sl:artist adele ] ], and the predicted sequence is: [in:playmusic [sl:year eighties m ] [sl:artist adele ] ], then the frame accuracy would be zero while the semantic accuracy would be one, since removing the slot values makes the two sequences identical. Since often times the model copies over extraneous characters for a slot value, semantic accuracy provides a measure of semantic understanding of the input that is robust to these errors. For Overnight we consider the logical form exact match accuracy as was done in Damonte et al. (2019).

Even though we motivate LORAS for structured prediction problems, for ATIS and SNIPS which only contain flat (non-hierarchical) frames, we evaluate vanilla label smoothing and LORAS on top of a BERT-based joint intent and slot tagging model (Chen et al., 2019). Chen et al. (2019) do not use any label smoothing and report state-of-the-art results for ATIS and SNIPs. For the BERT-based joint intent and slot tagging model we added label smoothing (vanilla and LORAS) only for the slot tagging component since the number of intents were too small for label smoothing to be useful. We ran each model three times with three different random initialization and report the test set exact match (EM) accuracy for the model that had the best validation set EM performance. Table 1 shows the performance of our method on ATIS and SNIPS data sets. LORAS consistently out-performs vanilla label smoothing and training with hard targets (no label smoothing) in almost all cases, while using label smoothing sometimes results in poorer performance over using hard targets. Since the ATIS data set contains simple, non-hierarchical queries, we observe that with BART the SA of the models is pretty close to the FA. On SNIPS we observe that the BART based models achieve almost perfect SA score with LORAS further improving SA by 0.3% over standard LS. We also observe a 1% drop in performance in using standard label smoothing over using hard targets. As we will show in the subsequent section, uniformly smoothing out the labels hurts performance of the models on

| Method | Reminder | | Weather | | Average | |
|---|---|---|---|---|---|---|
| | FA | SA | FA | SA | FA | SA |
| BART + Reptile[*] | **70.5** | **76.1** | 77.7 | 80.7 | **74.1** | 78.4 |
| BART + No LS | 69.8 | 75.7 | 75.1 | 78.8 | 72.5 | 77.3 |
| BART + LS | 67.6 | 72.6 | 76.6 | 79.9 | 72.1 | 76.3 |
| BART + LORAS | 70.1 | 75.7 | **78.1** | **81.4** | **74.1** | **78.6** |

Table 2: Test set frame accuracy (FA) and semantic accuracy (SA) on TOPv2 data set. Numbers in bold represent the best performing method for a given metric. *(Chen et al., 2020)

| Domain | Damonte et al. (2019) (Parsing) | No LS (seq2seq) | LS (seq2seq) | LORAS (seq2seq) |
|---|---|---|---|---|
| Publications | 40.4 | 64.0 | 64.0 | **66.5** |
| Calendar | 48.2 | **65.5** | 61.9 | 63.1 |
| Housing | 38.1 | 56.6 | 60.9 | **62.4** |
| Recipes | 63.0 | **80.6** | 80.1 | 78.7 |
| Avg. | 47.4 | 66.7 | 66.7 | **67.7** |

Table 3: Logical form exact match accuracy on target domains in the *Overnight* data set (Wang et al., 2015). The numbers for the neural transition-based parser of Damonte et al. (2019) are obtained from their paper (Table 3).

parts of the logical form for which the model is fairly accurate on. Lastly, from the results for the slot tagging we see that LORAS helps even in the multi-class classification setting [2]. To further investigate if the improvements come from LORAS discovering meaningful structure in the label space and not due to chance, we visualize the noise distributions learned by LORAS on the SNIPS data set and observe that it learns very distinct noise distributions for B and I tags (see Appendix B).

Table 2 shows the few-shot performance of LORAS on the TOPv2 data set and Table 4 shows the results on Overnight. For TOPv2, we also include the state-of-the-art BART+Reptile method (Chen et al., 2020) for comparison. On TOPv2, we see that LORAS comfortably out-performs label smoothing on both target domains. On the reminder domain, which contains logical forms with many hierarchical and nested structures, LORAS improves by 2.5%. Note that using label smoothing hurts performance as compared to training with hard (true) targets. Müller et al. (2019) also report that using label smoothing hurts knowledge distillation performance. What is surprising is that the performance of our simple LORAS technique can match the performance of the sophisticated meta-learning method in Chen et al. (2020) which is specifically designed for domain adaptation. For the more complex *reminder* domain with many nested structures in the parse tree, we observe that label smoothing degrades performance by as much as 3.1% as compared to LORAS. Finally, we also observe that, overall, LORAS provides the highest SA performance among all methods. On Overnight, LORAS improves upon both vanilla label smoothing and no label smoothing by 1% on an average and on the housing domain, which contains the most complex logical forms, LORAS improves by around 6% over the model with no smoothing. Note that our seq2seq model significantly outperforms (avg. improvement around 20%) the previous state-of-the-art on Overnight data set (Damonte et al., 2019).

**Exploring the structure recovered by LORAS.** In this section, we explore the structure in the label space that is recovered by LORAS in the TOPv2 and Overnight data sets by looking at the matrix $\mathbf{S} = \mathbf{L}\mathbf{L}^\top$ learned from data. Since the matrix $\mathbf{S}$ is larger than $50k \times 50k$, we visualize an informative sub-matrix in Figure 2. For each target token $k$, we visualize the noise distribution $\mathrm{softmax}(\mathbf{S}_{k,*})$ over a subset of vocabulary tokens. To select the most informative set of target tokens, we select the top 10 target tokens with lowest noise distribution entropy, and 5 target tokens with the highest noise distribution entropy. These noise distributions are shown in the top 10 and bottom 5 rows of Figure 2 respectively. Let the set $I$ denote these 15 target tokens, Figure 2 visualizes $(\mathrm{softmax}(\mathbf{S}))_{I,I}$, where the $\mathrm{softmax}$ is taken over the second dimension of $\mathbf{S}$.

---

[2]Our numbers for the slot tagging model without smoothing is slightly lower that those reported in Chen et al. (2019). We suspect this is because of our use of a validation set to select the best model initialization rather than directly selecting the model with the highest test set numbers. We use this procedure for all our models.

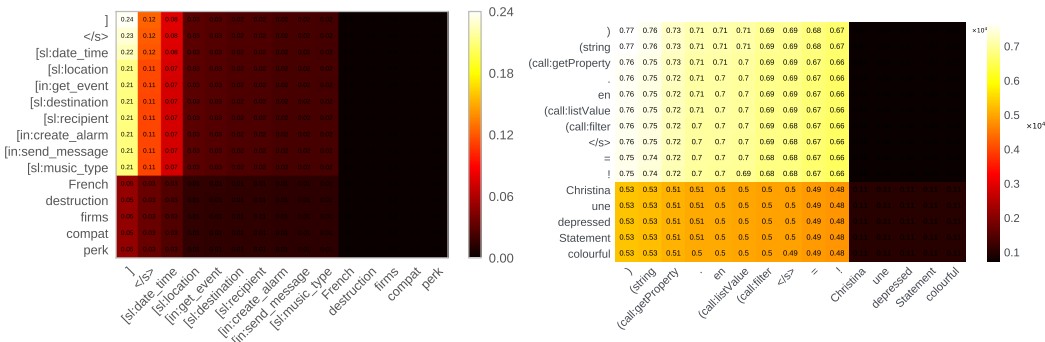

Figure 2: A sub-matrix of the noise distribution matrix learned from the target domains on: **(left)** the TOPv2 data in the few-shot setting, **(right)** the Overnight data set in the transfer learning setting. Row labels indicate the target token, and each row shows the noise distribution over a subset of the vocabulary tokens (column labels) given the target token. (Best viewed in color.)

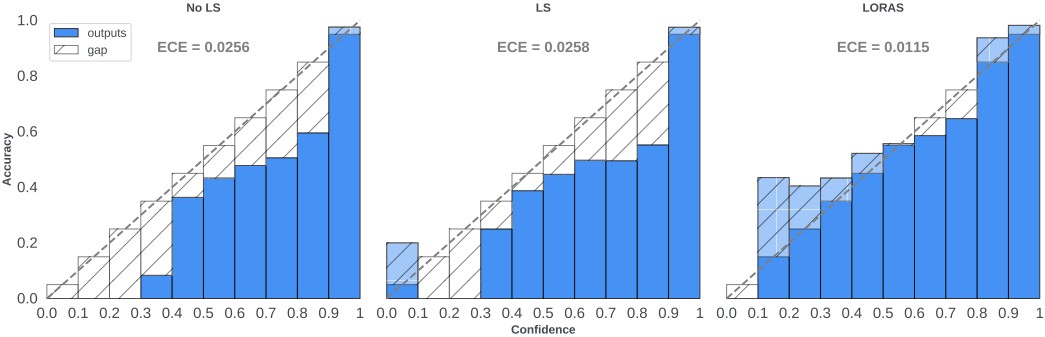

Figure 3: Calibration plots for model trained on hard targets (no label smoothing), standard label smoothing, and LORAS on the test set of target domains in the TOPv2 data set.

From Figure 2 we immediately observe that the noise distributions with the highest entropy correspond to utterance tokens and those with the lowest entropy correspond to ontology or special tokens like end of sentence (``). While the noise distribution corresponding to utterance target tokens are close to uniform, the noise distribution for ontology tokens places most of the mass on two tokens: `]` and ``. Thus as expected, LORAS groups the vocabulary into distinct groups: ontology and special tokens, and utterance tokens.

Further, among the non-utterance tokens we observe three distinct noise distributions corresponding to special tokens `]`, ``, and the rest of the ontology tokens. The noise distributions are also pretty informative, for instance, when decoding, the model has to decide between terminating a logical form with a closing bracket or a end-of-sentence tag. Therefore, these two tokens are most similar as is reflected in the noise distribution. Due to using rank parameter of 25 in experiments, we are not able to further discern patterns within the ontology tokens. However, it must be noted that learning more finer-grained relationships between labels would come at the cost of higher sample complexity.

To summarize, the low rank constraint coupled with the entropy constraint, helped discover relationships between ontology tokens and special tokens while learning an almost uniform noise distribution over the utterance tokens.

**Model calibration with LORAS.** In this section we dive into the issue of model calibration under no label smoothing, standard label smoothing, and LORAS. Intuitively, a probabilistic model is calibrated if the model's posterior probabilities are aligned with the ground-truth correctness likelihood (Guo et al., 2017). We use the framework of (Guo et al., 2017) to measure (mis-)calibration of the models using the expected calibration error. Label smoothing has been shown to improve calibration in image classification and machine translation (Müller et al., 2019), and on text classification using pre-trained transformer models (Desai & Durrett, 2020). We evaluate the test set model calibration

in the few-shot semantic parsing setting on the TOPv2 data set. Similar to (Müller et al., 2019), we evaluate calibration of the conditional probabilities $p(y_t = k \mid x, y_{1:t-1})$ on the test set of combined target domains on those examples where the predicted prefix $y_{1:t-1}$ is correct. Figure 3 shows the calibration plots along with the expected calibration error. We see that label smoothing does not improve model calibration in the structured prediction setting while LORAS produces fairly well-calibrated model reducing the expected calibration error by almost half. This is due to the fact that label smoothing makes the model *equally uncertain* about all tokens including top-level intents, closing brackets among others. While as observed in Rongali et al. (2020) and in our experiments, seq2seq models are good at learning the grammar of the parse trees, i.e., they produce well formatted trees with balanced bracket almost always. So the models should naturally be more confident about the location of `` and `]` tokens, while being less confident about novel intents and slots in the target domain. However, by uniformly perturbing the true targets, label smoothing makes the model equally uncertain about all kinds of target tokens thereby hurting calibration.

## 7 RELATED WORK

The main idea behind LORAS is to mix a non-uniform distribution, that is learned jointly with the model parameters, with the one-hot encoding of the true target to compute smooth labels or soft targets. Mixing non-uniform distributions with the one-hot encoding of the true target has been explored in different contexts using various approaches in the computer vision literature.

In the context of addressing overconfident predictions of deep neural networks Guo et al. (2017); Pereyra et al. (2017) propose penalizing the models output distributions using a hinge loss. While Reed et al. (2014) propose mixing the models predicted class probabilities with the true targets to generate soft targets to train deep networks on noisy labels. Lastly, knowledge distillation (Hinton et al., 2015), which involves using the output probabilities of a teacher model to train a student model, is another way in which a model (student) is trained with smooth labels. However, none of these approaches exploit structure in the label space. As opposed to the aforementioned papers, our approach involves learning a noise distribution parameterized by a low-rank symmetric similarity matrix. We then penalize the entropy of the noise distribution rather than the model's output distribution. As a result our approach scales to large label spaces with size exceeding fifty thousand. A part of our loss function is similar to that used in Bertinetto et al. (2020), however in Bertinetto et al. (2020) the authors assume a known hierarchical structure on the label space while we do not make such assumptions.

Our method is also superficially similar to label embedding approaches (Akata et al., 2015) which involves computing embeddings of labels and learning a mapping or compatibility function between input representation and the label embedding. Some notable work in this area is that of Frome et al. (2013) who initialize the label embeddings from a langauge model trained on data from Wikipedia. Xian et al. (2016) use different label embeddings like Glove and word2vec and learn a bilinear map between input image embeddings and class embeddings for zero-shot classification. In the NLP literature, Wang et al. (2018) compute label embeddings by using pre-trained word embeddings of the words in the labels and propose an attention mechanism to compute compatibility between input word embeddings and the label embeddings. However, unlike our method, these label embeddings are a part of the model and are needed during inference. Due to this limitation, such label embeddings cannot be readily incorporated into any existing architecture. Furthermore, in structured prediction problems where the label space contains special or abstract tokens, initiating good embedding matrices is not a trivial task.

## 8 CONCLUSION AND FUTURE WORK

In this paper, we developed a novel extension of label smoothing called low-rank adaptive label smoothing (LORAS) which improves accuracy by automatically adapting to the latent structure in the label space of semantic parsing tasks. While we evaluated LORAS on semantic parsing tasks and a slot tagging task, we believe that it will be useful for other seq2seq and multi-class classification tasks over structured label spaces and can always be used in place of standard label smoothing since it strictly generalizes the latter. It would also be interesting to initialize (part of) the embedding matrix used for computing the noise distribution using pre-trained models.

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

# A  APPENDIX

## A.1  LABEL SMOOTHING LOSS REFORMULATION

Starting with (1), representing the outputs $y$ as one-hot vectors, and representing the classification layer weights as a matrix $\mathbf{W} \in \mathbb{R}^{K \times d}$, we have the following label smoothing loss for the multi-class classification setting:

$$L(S, \mathbf{W}) = -(1 - \alpha) \sum_{(x,y) \in S} \log p(y \mid x, \mathbf{W}) - \alpha \sum_{y'} n(y') \log p(y' \mid x, \mathbf{W}),$$

where $n(\cdot)$ is the noise distribution, which is uniform for standard label smoothing. Writing the noise distributions as a vector $\mathbf{n} = (n(i))_{i=1}^{K}$, denoting $\overline{\phi} = \sum_{x \in S} \phi(x)$, and with $p(y \mid x, \mathbf{W}) =$

$\exp(y^\top \mathbf{W}\phi(x))/Z(\mathbf{W},x)$ where $Z(\mathbf{W},x) = \sum_{y'} \exp(y'^\top \mathbf{W}\phi(x))$ we have:

$$
\begin{aligned}
L(S,\mathbf{W}) &= -(1-\alpha)\sum_{(x,y)\in S}\{y^\top \mathbf{W}\phi(x) - \log Z(\mathbf{W},x)\} \\
&\quad - \alpha \sum_{y'} n(y')\left\{y'^\top \mathbf{W}\phi(x) - \log Z(\mathbf{W},x)\right\} \\
&= \sum_{(x,y)\in S} -(1-\alpha)y^\top \mathbf{W}\phi(x) + \log Z(\mathbf{W},x) - \alpha \sum_{(x,y)\in S}\mathbf{n}^\top \mathbf{W}\phi(x) \\
&= \sum_{(x,y)\in S} -(1-\alpha)y^\top \mathbf{W}\phi(x) + \log Z(\mathbf{W},x) \\
&\quad + \frac{\alpha}{2}\{\|\mathbf{n} - \mathbf{W}\overline{\phi}\|_2^2 - \|\mathbf{n}\|_2^2 - \|\mathbf{W}\overline{\phi}\|_2^2\}.
\end{aligned}
$$

Denoting the re-scaled negative log-likelihood by $l(x,y;\mathbf{W},\alpha) = -(1-\alpha)y^\top \mathbf{W}\phi(x) + \log Z(\mathbf{W},x)$, and since $\|\mathbf{n}\|_2$ is a constant term for minimizing $L(S,\mathbf{W})$ with respect to $\mathbf{W}$, and dropping the term $\|\mathbf{W}\overline{\phi}\|_2^2$, we arrive at the following upper bound on the label smoothing loss:

$$
\widetilde{L}(S,\mathbf{W}) = \sum_{(x,y)\in S} l(x,y;\mathbf{W},\alpha) + \frac{\alpha}{2}\|\mathbf{n} - \mathbf{W}\overline{\phi}\|_2^2 \tag{6}
$$

## A.2 PAC-Bayesian generalization bound proof

We re-state the theorem for convenience.

**Theorem 2** (PAC-Bayesian generalization bound). *Set the distribution $Q(\mathbf{W})$, parameterized by $\mathbf{W}$ with bounded induced norm, over the weights $\mathbf{W}'$ to be such that each column $\mathbf{W}'_{*,i}$ is sampled i.i.d. from the Gaussian distribution $\mathcal{N}(\mathbf{W}\overline{\phi}, \mathbf{I})$. If $\alpha = 2d/\sqrt{N}$, where $N = |S|$ is the number of samples, then with probability at least $1 - \delta$ the generalization error is bounded as follows:*

$$
\overline{R}(Q(\widehat{\mathbf{W}})) - \overline{R}(Q(\overline{\mathbf{W}})) \le \frac{2d}{\sqrt{N}}\|\mathbf{n} - \overline{\mathbf{W}}\overline{\phi}\|_2^2 + \frac{1}{\sqrt{N}}\log \frac{2e^{\frac{b^2}{8}}}{\delta}.
$$

*Proof.* Denote $l(S;\mathbf{W},\alpha) = \sum_{(x,y)\in S} l(x,y;\mathbf{W},\alpha)$ as the (re-scaled) negative log-likelihood of the data $S$. Let $\widehat{R}(Q(\mathbf{W})) = \mathbb{E}_{\mathbf{W}'\sim Q(\mathbf{W})}[l(S;\mathbf{W}',\alpha)]$ be the empirical negative log-likelihood. Let $\overline{R}(Q(\mathbf{W})) = \mathbb{E}_{\mathbf{W}'\sim Q(\mathbf{W})}\left[\mathbb{E}_{(x,y)}[l(x,y;\mathbf{W}',\alpha)]\right]$ be the expected negative log-likelihood. Choose the prior distribution to be the following product distribution parameterized by the noise distribution: $P = \prod_{i=1}^{d} \mathcal{N}(\mathbf{n}, \mathbf{I})$. Since $Q(\mathbf{W}) = \prod_{i=1}^{d} \mathcal{N}(\mathbf{W}\overline{\phi}, \mathbf{I})$, we have that

$$
\begin{aligned}
\mathbb{KL}\left(Q(\mathbf{W})\|P\right) &= \sum_{i=1}^{d} \mathbb{KL}\left(\mathcal{N}(\mathbf{W}\overline{\phi}, \mathbf{I})\big\|\mathcal{N}(\mathbf{n}, \mathbf{I})\right) \\
&= d\|\mathbf{n} - \mathbf{W}\overline{\phi}\|_2^2.
\end{aligned}
$$

From the PAC-Bayesian theorem (McAllester, 2003) we have that with probability at least $1 - \delta$, $\varepsilon = (1/\sqrt{N})\log 2e^{b^2/8}/\delta$ and for all $\mathbf{W} \in \mathbb{R}^{K\times d}$ with bounded induced norm:

$$
\begin{aligned}
\left|\widehat{R}(Q(\mathbf{W})) - \overline{R}(Q(\mathbf{W}))\right| &\le \frac{1}{\sqrt{N}}\{\mathbb{KL}\left(Q(\mathbf{W})\|P\right) + \varepsilon\} \\
&= \frac{1}{\sqrt{N}}\left\{\|\mathbf{n} - \mathbf{W}\overline{\phi}\|_2^2 + \varepsilon\right\}, \tag{7}
\end{aligned}
$$

where $b$ is the Lipschitz constant of the loss $l$ which is bounded under our assumption that $\|\phi(\cdot)\|_2 \le 1$ and $\mathbf{W}$ have bounded induced norm. Setting $\alpha = 2d/\sqrt{N}$ and since $\widehat{\mathbf{W}} \in \arg\min_{\mathbf{W}} \widehat{R}(Q(\mathbf{W})) + d/\sqrt{N}\|\mathbf{n} - \mathbf{W}\overline{\phi}\|_2^2$ we have that

$$
\widehat{R}(Q(\widehat{\mathbf{W}})) + d/\sqrt{N}\|\mathbf{n} - \widehat{\mathbf{W}}\overline{\phi}\|_2^2 \le \widehat{R}(Q(\overline{\mathbf{W}})) + d/\sqrt{N}\|\mathbf{n} - \overline{\mathbf{W}}\overline{\phi}\|_2^2 \tag{8}
$$

Therefore, from (7) and (8) we have that:

$$
\begin{aligned}
\overline{R}(Q(\widehat{\mathbf{W}})) - \overline{R}(Q(\overline{\mathbf{W}})) &\leq \widehat{R}(Q(\widehat{\mathbf{W}})) + d/\sqrt{N} \left\| \mathbf{n} - \widehat{\mathbf{W}}\phi \right\|_2^2 \\
&\quad - \widehat{R}(Q(\overline{\mathbf{W}})) + d/\sqrt{N} \left\| \mathbf{n} - \overline{\mathbf{W}}\overline{\phi} \right\|_2^2 + 2\varepsilon \\
&\leq \widehat{R}(Q(\overline{\mathbf{W}})) + d/\sqrt{N} \left\| \mathbf{n} - \overline{\mathbf{W}}\overline{\phi} \right\|_2^2 \\
&\quad - \widehat{R}(Q(\overline{\mathbf{W}})) + d/\sqrt{N} \left\| \mathbf{n} - \overline{\mathbf{W}}\overline{\phi} \right\|_2^2 + 2\varepsilon \\
&= 2d/\sqrt{N} \left\| \mathbf{n} - \overline{\mathbf{W}}\overline{\phi} \right\|_2^2 + 2\varepsilon
\end{aligned}
$$

$\square$

## A.3 Proof of Proposition 1

We re-state the proposition for convinience.

**Proposition 2.** *Setting $q = \eta = 0$ and $\mathbf{L} = \mathbf{1}$, where $\mathbf{1}$ is the $K$-dimensional vector of ones, $L^{\mathrm{LORAS}}(S) = L^{LS}(S) = \sum_{(x,y)\in S} \sum_{t=1}^{|y|} H(\mathbf{y}_t^{LS}, \mathbf{p}_t)$.*

*Proof of Proposition 1.* With $q = \eta = 0$ we have that

$$
L^{\mathrm{LORAS}}(S) = \sum_{(x,y)\in S} \sum_{t=1}^{|y|} H(\mathbf{y}_t^{\mathrm{LORAS}}, \mathbf{p}_t)
$$

With $\mathbf{L} = \mathbf{1}$ we have that $\mathbf{L}_{y_t,*}(\mathbf{L}_{j,*})^\top = 1, \forall y_t, j$. Therefore, $H(\mathbf{y}_t^{\mathrm{LORAS}}, \mathbf{p}_t) = H(\mathbf{y}_t^{LS}, \mathbf{p}_t)$. $\square$

Since setting $q = \eta = 0$, and the rank parameter $r = 1$, we have that $\mathbf{L} = \mathbf{1}$ is in the solution path of minimizing the LORAS loss given in (4). Therefore, adaptive label smoothing strictly generalizes label smoothing.

## A.4 Experiment Details

| Domains | Avg. depth | # Train examples |
|---|---|---|
| Blocks | 4.64 | 1596 |
| Social network | 5.52 | 3535 |
| Basketball | 5.83 | 1561 |
| Restaurants | 4.65 | 1325 |
| All | | 8017 |
| Housing | 4.60 | 752 |
| Calendar | 4.48 | 669 |
| Publications | 4.50 | 640 |
| Recipes | 4.32 | 864 |
| All | | 2925 |

Table 4: Statistics of the Overnight data set (Wang et al., 2015). The top four domains are chosen as source domains while the bottom four are chosen as target domains. Average depth is the average of the maximum depth of logical forms (trees) in the test set.

**Overnight data set.** Statistics of the Overnight data set (Wang et al., 2015) is shown in Table 4. For our transfer learning setting we chose the four smallest domains as target domains and the rest as source domains. As was done in prior work (Wang et al. (2015), Damonte et al. (2019)) we randomly select 20% of training data from each domain as the validation set which is used for model selection and hyper parameter tuning. We used the test set provided by the data set authors (Wang et al., 2015) for reporting the final performance of all the models.

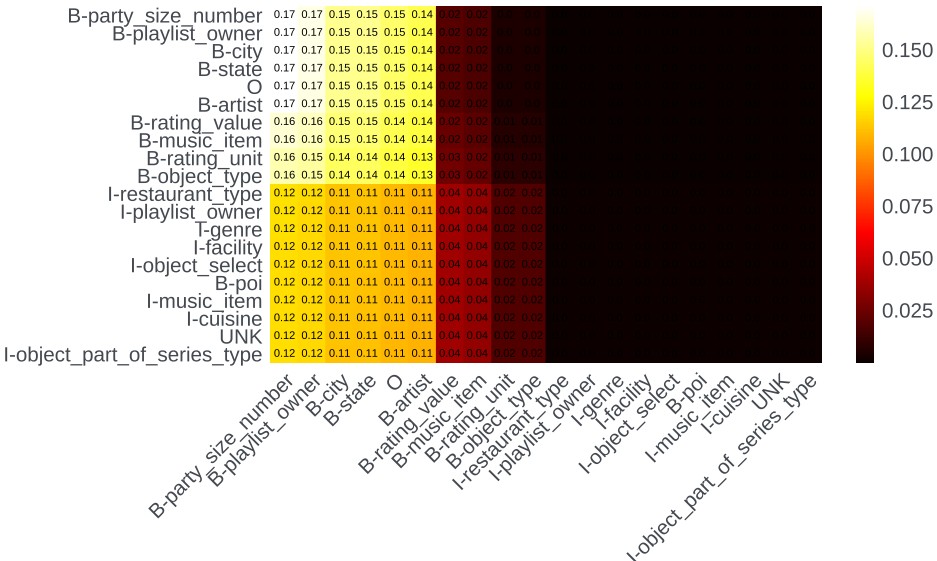

Figure 4: A sub-matrix of noise distribution matrix learned over the SNIPS data set. The top ten rows shows labels with the lowest entropy noise distribution while the bottom ten rows shows labels with the highest entropy noise distributions.

After some minor pre-processing, an example utterance and it's corresponding logical form from the data is shown below:

*Utterance:* meetings that start later than the weekly standup meeting
*Logical form:*
```
(call:listValue (call:filter (call:getProperty (call:singleton
en.meeting ) (string !type ) ) (call:ensureNumericProperty
(string start_time ) ) (string > ) (call:ensureNumericEntity
(call:getProperty en.meeting.weekly_standup (string end_time )
) ) ) )
```

We add the function names like `(call:listValue`, variable types like `(string`, and closing bracket `)` at the end of the GPT2 vocabulary as ontology tokens. The final vocabulary size was 50284.

**Model details.**    The model from Rongali et al. (2020) comprises of a seq2seq Transformer model and a Pointer Generator Network (See et al., 2017). When using RoBERTa, only the encoder is initialized with the pre-trained embeddings while a 3-layer 256-hidden-unit decoder is trained from scratch. Unlike RoBERTa, BART provides both a pre-trained encoder and decoder which makes it an ideal option for initializing seq2seq models. Lastly, in our implementation, the model generates ontology tokens from the vocabulary while the slot values are always copied over from the utterance using the copy mechanism. For ATIS and SNIPS we did not canonicalize the data (Chen et al., 2020), i.e. alphabetically order the slots at the same level, during pre-processing.

**Training details.**    For all our experiments we used the following early stopping criteria: we stop training after the validation accuracy does not improve for 20 epochs. All the models were trained on Nvidia Telsa GPUs with 16GB of RAM. For ATIS and SNIPS, the models were trained on a single GPU while for TOPv2 models were trained on 2 GPUs. We used Adam optimizer with default settings, inverse square root learning rate schedule, and a batch size of 32 for all our experiments.

## B   STRUCTURE RECOVERED BY LORAS IN THE SNIPS DATA SET

Similar to Figure 2, Figure 4 visualizes the noise distributions of labels with the lowest entropy (top ten rows) and the those with the highest entropy (bottom ten rows). We observe that top-10 low entropy noise distributions correspond to mostly "B" tags (beginning of slot) while top-10 high entropy noise distributions (close to uniform) are learned for "I" tags. This is intuitive, since the model is more likely to confuse one B tag with another B tag (e.g. B-city vs B-state) rather than one B tag with another I tag. Therefore, by smoothing over closely related labels (tags) LORAS is able to force the model to learn better representations.

