# OpenReview forum: "Learning Better Structured Representations Using Low-rank Adaptive Label Smoothing"
_ICLR.cc/2021/Conference — ICLR 2021 Poster_

### Official Review · AnonReviewer4 · 2020-10-28
**Official Blind Review #4**

**Rating:** 7
**Confidence:** 4

**Review:**

This paper theoretically analyzes "label smoothing” (LS)  with PAC-Bayesian bound and motivated from their analysis, proposes a new method: LORAS.  In their theoretical analysis, they identify that the generalization error depends on the smoothing distribution. So they propose to learn the smoothing distribution in LORAS. In doing so, to overcome the computational issues & overfitting to diagonal smoothing, they propose Low-rank approach which seems to be successful. The authors show experimental results on semantic parsing dataset (ATIS, SNIPS, and TOPv2) where the LORAS shows better performance than no LS, LS, and some other SOTA model (with one exception of LORAS  vs. no LS on ATIS)

**Quality**:
* The writings and organizations of the paper are very clear. Didn’t go into the details of proof steps in the appendix, but seems right.
* This analysis sets up clear motivations for proposing new method: LORAS.
* The experiment results are convincing. It might have been even more convincing if LORAS was applied to other tasks to show that it generally works.

**Clarity**:
I think most parts of the paper was clearly written. Addressing the following questions might paper even more clearer:
* In section 3, if the authors could simply write the analytical form of probability $p_{t,j}$ as written in Appendix A.1, I think the context of how  $\lvert\lvert n-W \bar\phi\rvert\rvert$ came in eq.(2) will be clearer.
* [suggestion] When expressing absolute changes in accuracy, perhaps say 2 percent point to differentiate with accuracy performing 2 percent better? (e.g. I am assuming calibration error became 55% smaller (almost half) rather than 55 percent point?)
* In section 6. Results, the paper says that "LORAS consistently out-performs" vanilla LS & hard targets in “all cases”, but isn’t SA worse than (No LS, LS) in case of RoBERTa & ATIS combination in Table 1? I wasn’t sure whether this was the wrong description or whether the table numbers were wrong. Please fix this later.
* For Figure 3, ALS on the rightmost figure should become LORAS?

Significance
* Pros
    * The deliveries were clear where I was able to see the motivation of this work: that generalization error (on the upper bound of loss) is bounded by the difference of $n$ and $W\bar\phi$. And this also seems to be a contribution of this work.
    * The low-rank approximation seems like a clever way to resolve both computational issues & overfitting to diagonal matrix.
    * Experimental results seem to support the success of the proposed method.
* Cons
    * This is not a big complaint, but if the paper could have provided strong experiments with other tasks as well, then I think the paper would have become stronger.
    * Likewise, it would have been great if the authors could have tested on much larger label size (and much smaller label size to check as well) to see whether LORAS can actually handle very large label space as claimed.

---

> ### Author Response · Authors · 2020-11-24
> **Responses to specific comments**
>
> > The experiment results are convincing. It might have been even more convincing if LORAS was applied to other tasks to show that it generally works.
>
> Thank you for your feedback. We have now added more experiments. See "general comments".
>
> > "LORAS consistently out-performs" vanilla LS & hard targets in “all cases”, but isn’t SA worse than (No LS, LS) in case of RoBERTa & ATIS combination in Table 1? For Figure 3, ALS on the rightmost figure should become LORAS?
>
> We have changed the wording to "almost all cases" and changed the label of figure to LORAS.
>
> > could have tested on much larger label size (and much smaller label size to check as well) to see whether LORAS can actually handle very large label space as claimed.
>
> Note that the vocab size in our experiments when using BART is larger than 50k (GPT2 vocab + ontology tokens). The smoothing matrix that we learn is larger than 50k x 25 (where 25 is the rank parameter used in experiments) and in the few-shot and transfer learning experiments we learn this from a few hundred samples per domain. We have now also evaluated LORAS on top of the BERT based slot tagging model for ATIS and SNIPS where the vocab size (for slot prediction) is 122 and 74 respectively.

---

### Official Review · AnonReviewer1 · 2020-10-28
**Boost the parsing through low-rank label smoothing**

**Rating:** 6
**Confidence:** 4

**Review:**

The paper proposes a label-smoothing method upon the low-rank assumption of the output dimension, especially when the output dimension is large. The contribution of this work is two folds: first, highlighted the importance of informative label smoothing through better bound, and second, proposed one label smoothing with low-rank assumption. It is overall a good paper but there are a few concerns:

1. I didn't go through the theoretic proof in detail but it is not obvious how this theoretic motivation is correlated with the low-rank assumption. It will be better for the authors to highlight more about the theoretic result and the empirical algorithms.

2. The experiments are only conducted on parsing tasks, but the authors have claimed the possibility in other areas. I would suggest the authors conduct one/two more experiments on other tasks, like language modeling with large vocab. If additional experiments are presented, I will increase my score.

3. Low-rank output dimension is a common assumption in many papers and it seems there are some mission citations. Few examples are listed below:

https://arxiv.org/abs/1711.03953
http://papers.nips.cc/paper/9723-mixtape-breaking-the-softmax-bottleneck-efficiently
http://papers.nips.cc/paper/7312-sigsoftmax-reanalysis-of-the-softmax-bottleneck

4. In most cases, the low-rank assumption is applied when the vocab size is huge. However, the proposed method requires to have to NxN matrix - S to calculate the L. If the N is 1M, like in language modeling, will it be a bottleneck here?

---

> ### Author Response · Authors · 2020-11-24
> **Responses to specific comments**
>
> > but it is not obvious how this theoretic motivation is correlated with the low-rank assumption.
>
> We would like to clarify that the low-rank assumption is not motivated from the theoretical analysis but provides a way to learn smoothing distributions over large vocabs. Our theoretical results motivate using more structured smoothing distributions to achieve better generalization and LORAS provides a specific recipe for achieving that.
>
> > I would suggest the authors conduct one/two more experiments on other tasks, like language modeling with large vocab.
>
> We have now added results for Overnight which is a structured prediction task and also evaluated our approach on top of slot tagging model (which is a sequence tagging task) for ATIS and SNIPs. We believe LM might not be an appropriate task since the label space doesn't appear to have obvious structure.
>
> > In most cases, the low-rank assumption is applied when the vocab size is huge. However, the proposed method requires to have to NxN matrix - S to calculate the L. If the N is 1M, like in language modeling, will it be a bottleneck here?
>
> We would like to clarify that we do not compute the S matrix directly but work with the L matrix which is N x r where N is the vocab size and r is the rank parameter. When using BART for ATIS, SNIPS, TOPv2, and Overnight, our vocab is the GPT2 vocab plus the ontology tokens which makes the vocab size larger than 50k. As pointed in the paper the memory requirement of our method scales linearly with the vocab. Current LMs use the same BPE vocab. However, we can imagine learning smoothing distributions for a subset of the vocab (e.g. those corresponding to ontology tokens) can also be useful since our analysis reveals that LORAS ends up learning a smoothing distribution that is close to uniform for non-ontology tokens.

---

### Official Review · AnonReviewer3 · 2020-10-30
**Sound approach but not very convincing experiments**

**Rating:** 6
**Confidence:** 2

**Review:**

The paper proposes LORAS (low-rank adaptive label smoothing) that learns with soft targets for better generalizing to the latent structure in the label space, and the experiments on three semantic parsing datasets show the improvement over the no smoothing methods.

The idea is intuitive and reasonable, and the proposed method and proof seem sound.
Also, as mentioned in the paper, the approach can better consider the latent structure in the label space and is more suitable for structure prediction tasks.

However, there are some concerns about the experiments, so that it is not very convincing about the effectiveness of the proposed method.
The conducted experiments include three datasets, ATIS, SNIPS, and TOPv2, all of which are about semantic parsing/natural language understanding.
This paper formulates the task as a seq2seq task, where the output sequence contains the intent and slot-value information, but most prior work for ATIS and SNIPS formulated the task as classification and tagging problems and the achieved frame accuracy is higher than the performance shown in this paper.
For example, simply using BERT for joint training intent classification and slot filling (https://arxiv.org/pdf/1902.10909.pdf) achieved the frame accuracy of 88.2 in ATIS and 92.8 in SNIPS, which are better than the scores using LORAS reported in Table 1.
Because the proposed label smoothing method can be utilized for not only sequence generation tasks but also classification tasks, it would be more convincing if the results can be directly compared with the prior work.
Authors are suggested add the proposed LORAS on the existing SOTA models in order to better convince the readers.

Another concern is about the evaluation metric used in the paper. Frame accuracy is common for the task, but semantic accuracy seems not to be used in the prior work.
From my perspective, evaluating the performance based on semantic accuracy simplifies the task, and the classification models should easily achieve better performance compared to the generation based methods.
This also implies that the authors should perform classification models in addition to generation-based models.

Moreover, in TOPv2, the paper adds LORAS on BART and show the small improvement and mentions that the results are comparable to the meta-learning method designed for domain adaptation. Is it possible to add the proposed LORAS above the model proposed by Chen et al. (2020) and further improve the performance?

Another issue to be addressed is that the datasets this paper uses are not well-known structure prediction task, because ATIS and SNIPS only contain very flat structures for the semantic frames, which may not be suitable to demonstrate the effectiveness of the proposed method for structure prediction.
Other structure prediction datasets can be included in the experiments, and probably the improvement can be more significant due to the complex latent structure in the label space.

In sum, this paper proposes a sound method for label smoothing and claims that it can benefit the generalization capability based on learning the latent structure in the label space. The experiments on semantic parsing are not very convincing, because the results cannot directly compare with the prior work's or the improvement is relatively subtle.
To better align with the claim, structure prediction datasets should be considered in the paper.

After reading the responses and checking the additional experiments, I changed the score for this paper.

---

> ### Author Response · Authors · 2020-11-24
> **Responses to specific comments**
>
> > Authors are suggested add the proposed LORAS on the existing SOTA models in order to better convince the readers.
>
> Thank you for your suggestion. The main motivation for our work was structured prediction problems, but as suggested, we have now added results for slot tagging models.
>
> > evaluating the performance based on semantic accuracy simplifies the task, and the classification models should easily achieve better performance ****
>
> We use frame accuracy as the primary metric and use semantic accuracy (SA) to get insights into the structured prediction performance. We do not use SA alone. We would also like to point out that the TOPv2 data set has 166 ontology tokens and trees of depth 8 making the space of possible trees enormous.
>
> > Is it possible to add the proposed LORAS above the model proposed by Chen et al. (2020) and further improve the performance?
>
> This is an interesting question which we leave for future work since it is unclear if all the parameters, including the smoothing distribution, should be meta-learned.
>
> > datasets this paper uses are not well-known structure prediction task, because ATIS and SNIPS only contain very flat structures
>
> We have now added another structured prediction data set. On ATIS and SNIPS our BART initialized seq2seq model out-performs those of Rongali et al.

---

### Official Review · AnonReviewer2 · 2020-11-02

**Rating:** 6
**Confidence:** 5

**Review:**

This paper proposes to improve upon label smoothing (LS) by adapting the noise distribution used in LS from uniform to a distribution that better represents the correlation/similarity between the candidate space (types in vocabulary).

-- It proposes to learn the appropriate noise distribution (similarity matrix) during training. This matrix is parametrized via a low-rank approximation which prevents it from collapsing to a diagonal matrix and become ineffective.

-- This approach  introduces more hyperparameters and the tuning of these hyperparameters appears to be non-trivial

-- This approach is empirically compared to vanilla label smoothing and no LS on the task fo semantic parsing which has natural groups of types in the vocabulary.

-- The approach seems to be learning appropriate correlations among the vocabulary items (as seen in the visualization).

-- The improvement over LS is consistent but small on the two semantic parsing datasets. Additionally, uniform LS seems to be hurting performance in many cases when compared to no LS, hence the improvements of the proposed approach over no-LS are modest.

-- The utility of the proposed approach becomes apparent in the few-shot setting where the gains seem significant. More analysis of this would strengthen the paper.

-- The approach also seems to improve model calibration.

-- The theoretical analysis is reasonable but it is unclear about its contribution toward understanding the effect of label smoothing, mainly because of the assumption that the embeddings and most of the network is frozen except the last linear layer which is never the case during training of these models. It is a straightforward extension of analysis provided in prior work and is limited in terms of explaining the effect of the noise distribution/ proposed approach on the optimization of the models.

-- Another straightforward experiment would have been to use prior knowledge to hand-design the noise distribution. For example, a simple baseline would be: manually cluster types based upon their membership as an intent, slot , or a word and use uniform distributions within these groups. It would be interesting to compare how the learned similarity matrix is different form hand-designed prior matrices.

---

> ### Author Response · Authors · 2020-11-24
> **Responses to specific comments**
>
> > introduces more hyperparameters and the tuning of these hyperparameters appears to be non-trivial
>
> We do agree that our approach introduces 3 more hyper-parameters but through our experiments we show that we can get good improvements without significant tuning: we set $\eta$ to be 0.1 in all our experiments and didn't tune it. The label smoothing term can also be set to 0.2 (typically 0.1 is used in the literature). The only parameter that is important is the rank parameter but we observed improvements with multiple values of rank parameter.
>
> > More analysis of few shot setting.
>
> Thank you for the suggestion. We have now added transfer learning results on another data set together with an analysis of the results.
>
> > assumption that the embeddings and most of the network is frozen except the last linear layer
>
> We focus on this setting because Mueller et al. 2019 (When does label smoothing help) empirically argue that label smoothing impacts the last layer. We believe our PAC Bayesian analysis can be carried forward to the case when the embeddings are not fixed by putting priors over the embeddings (similar to Neyshabur et al 2018). This would lead to another KL divergence term in the generalization bound but it wont reveal any additional insights about the effect of label smoothing. Lastly, we would like to highlight that existing PAC Bayesian analysis of neural networks focus on margin losses, whereas we show that label smoothing introduces a regularization term that depends on the data, which introduces new challenges for theoretical analysis.
>
> > unclear about its contribution toward understanding the effect of label smoothing
>
> Our analysis reveals the role of the noise distribution in achieving good generalization performance — specifically that more informative noise distribution reduce the generalization error. Furthermore, our analysis reveals that "as the number of samples N → ∞, α → 0 and less smoothing of the hard targets is needed to achieve generalization". We actually observe this in practice where we see the most benefit of LORAS in low-resource settings.
>
> > prior knowledge to hand-design the noise distribution
>
> This is an interesting suggestion which needs further in-depth exploration. However from our visualization of the learned noise distributions for TOPv2 and Overnight, the most informative noise distributions are learned for ontology tokens like closing braces "]}, ")", special tokens like end of sentence "</s>", "." among others. Hand-designing noise distributions for these is non-trivial.

---

### Author Response · Authors · 2020-11-24
**General comments**

We thank all the reviewers for their comments and feedback for improving the paper. As suggested by reviewers, we have added results on a new data set: Overnight ("Building a semantic parser overnight", Wang et al) where the goal is to predict an executable logical form that can be executed against a database to answer a natural language query. We also evaluated our method on ATIS and SNIPS as a joint intent and slot tagging task ( in addition to the seq2seq formulation ) as suggested by R3. The results are summarized below:

## Overnight

1. Table 3 describes our results on Overnight in the transfer learning setting where we select the four largest domains as source domains and the four smallest domains as the target domains. LORAS improves over no label smoothing and vanilla label smoothing by 1% on an average over 4 target domains and on the "housing" domain the improvement is around 6%.
2. Figure 2 also includes a visualization of the noise distribution learned on Overnight and the same observations that we had for TOPv2 hold for Overnight as well where we observe distinct noise distributions for ontology and special tokens and other tokens.
3. We report significantly higher exact match accuracy of logical form than previous work (improvement around ~ 20%).

## Joint Intent prediction and slot tagging on ATIS and SNIPS

1. The last row of Table 1 shows the results of vanilla label smoothing and LORAS on top of the BERT based slot tagging model of Chen et al (2019) where we only use smoothing for the slot tagging component (and not the intent classification component since the number of intents is too small for smoothing to be useful). LORAS performs better than no label smoothing and vanilla label smoothing.
2. The numbers that we report on ATIS and SNIPS are slightly lower than those reported in Chen et al (2019). We give an explanation for this in footnote 2 in page 7. We further investigate if the improvements by LORAS are meaningful or due to chance. Towards that end we visualize the noise distribution learned by LORAS over SNIPS in Fig. 4 (Appendix, page 14). We observe that LORAS learns distinct noise distributions for "B"  and "I" tags and the noise distribution makes intuitive sense since the model is more likely to confuse one B tag with another B tag (e.g. B-city vs B-state) rather than one B tag with another I tag.

---

### Decision · Program_Chairs · 2021-01-07
**Final Decision**

**Decision:**

Accept (Poster)

**Comment:**

The paper proposes LORAS (low-rank adaptive label smoothing) for training with soft targets with the goal of improving performance and calibration for neural networks. The authors derive PAC-Bayesian generalization bounds for label smoothing and show that the generalization error depends on choice of the noise (smoothing) distribution. Empirical results demonstrate the effectiveness of the approach. All reviewers recommend acceptance.